# Alveolar Echinococcosis in a Patient with Presumed Autoimmune Hepatitis and Primary Sclerosing Cholangitis: An Unexpected Finding after Liver Transplantation

**DOI:** 10.3390/pathogens12010073

**Published:** 2023-01-03

**Authors:** Florian Fronhoffs, Leona Dold, Marijo Parčina, Arne Schneidewind, Maria Willis, Thomas F. E. Barth, Tobias J. Weismüller, Taotao Zhou, Philipp Lutz, Julian A. Luetkens, Peter Gerlach, Steffen Manekeller, Jörg C. Kalff, Tim O. Vilz, Christian P. Strassburg, Glen Kristiansen

**Affiliations:** 1Institute of Pathology, University Hospital Bonn (UKB), 53127 Bonn, Germany; 2Department of Internal Medicine I, University Hospital Bonn (UKB), 53127 Bonn, Germany; 3German Center for Infection Research (DZIF), Partner Site Cologne-Bonn, 53127 Bonn, Germany; 4Institute of Medical Microbiology, Immunology and Parasitology, University Hospital Bonn (UKB), 53127 Bonn, Germany; 5Department of General, Visceral-, Thoracic and Vascular Surgery, University Hospital Bonn (UKB), 53127 Bonn, Germany; 6Institute of Pathology, Ulm University, 89081 Ulm, Germany; 7Department of Radiology, University of Bonn (UKB), 53127 Bonn, Germany; 8Institute of Pathology, Bonn-Duisdorf, 53123 Bonn, Germany

**Keywords:** echinococcus multilocularis, alveolar echinococcosis, liver, transplantation, PSC

## Abstract

Primary sclerosing cholangitis is an important reason for liver transplantation. Hepatic alveolar echinococcosis (AE) is caused by *Echinococcus multilocularis* and presents characteristic calcified conglomerates detected by ultrasound or computed tomography scan of the liver. Symptoms of AE only occur after a long period of infection when cholestasis or cholangitis becomes apparent. Here, we report on a patient with presumed autoimmune hepatitis and primary sclerosing cholangitis. After liver transplantation, alveolar echinococcosis was diagnosed in the liver explant.

## 1. Case Report

A 43-year-old male from western Germany with a history of autoimmune hepatitis (AIH), diagnosed two years earlier, was referred to our department of hepatology for further evaluation. His general condition showed deterioration. During the last two years, he had lost 8 kg of weight and suffered from pruritus and fatigue. He reported dark urine and pale stool and showed scleral icterus. His medication consisted of 150 mg of azathioprine, 9 mg of budesonide and 1000 mg/d of ursodeoxycholic acid (UDCA).

Two years before presentation to our institution, a liver biopsy (Figure 2A) had been performed due to elevated liver enzymes, showing a chronic-active, predominantly portal and discrete lobular inflammatory infiltrate with low activity and few eosinophilic granulocytes. The inflammatory infiltrate involved the bile ducts only focally. The patient’s history was not indicatory of a drug-toxic genesis. In agreement with elevated ANA levels (titer 1:800), AIH was diagnosed, and immunosuppressive treatment was initiated. At that time, MRCP showed intrahepatic cholestasis of both liver lobes with accentuation in intrahepatic bile ducts, while calibers of extrahepatic bile ducts remained normal.

At first presentation to our department, no autoantibodies specific for AIH were detected (ANA, SLA/LP, SMA/Actin, LKM, or pANCA). Total IgG and IgG4 levels were within the normal range.

ERCP showed a dominant stricture of the liver hilus and advanced atrophy of the right sided biliary ducts with PSC typical irregularities, while the left sided bile ducts were almost normal and showed hypertrophy. A balloon dilation of the hiliary stenosis to the left hepatic duct was performed (Figure 1B). The histology of the common bile duct (DHC) showed partly biliary, partly intestinal cells and columnar epithelial mucosa without atypia. Additionally, sparse neutrophilic granulocytes were seen in the inflammatory exudate. Neither dysplasia nor plasmacellular inflammation was detected. A colonoscopy was performed, excluding an associated chronic inflammatory bowel disease. Viral infections by CMV, HSV, VZV, and EBV and hepatitis viruses A, B and C were serologically excluded. A CT scan and an MRI were performed (Figure 1D,E). Results were interpreted as consistent with chronic cholangitis due to PSC [1]. The patient was listed for liver transplantation with a diagnosis of advanced PSC and concomitant AIH. During follow-up, MRCP was performed half a year before transplantation, showing irregularly configured and dilated bile ducts, predominantly centrally located, with adjacent hyperperfused liver parenchyma in the right lobe of the liver (Figure 1C).

A diffusion restriction was detected, most probably due to a very small super-infected bilioma. Because of the small sizes of the lesions, lack of symptoms and stable CRP and cholestasis parameters, a diagnostic biopsy was not performed. Within the year before transplantation, several biopsies of the main bile ducts were obtained during ERCP. These congruently showed fibrosis and chronic, as well as granulocytic, inflammation. Two months before transplantation, a biopsy showed large amounts of IgG4-positive plasma cells, and PSC associated with IgG4 infiltration was assumed (Figure 2B). Clinically, the patient benefited from repeated ERCP interventions with mechanical dilatation of biliary strictures.

One and a half years after initial presentation to our institution, liver transplantation was performed using the piggy-back technique with cavocavostomy.

The explanted liver weighed 1204 g and measured 210 × 180 × 80 mm. The right lobe was atrophic and showed several retractions of the capsule. On sections (including the hilus region, Figure 3), multiple, rounded, abscess-like lesions up to 14 mm became visible, some of them containing greenish fluid, while others showed calcified rims.

In some areas, the adjacent liver tissue was fibrotic or hemorrhagic. The histology of the explanted liver revealed multiple metacestodes of *Echinococcus multilocularis* with fragments of the eosinophilic laminated layers (Figure 2C,D and Figure 4A,D. The amount of IgG4-positive plasma cells was strikingly increased (Figure 4F).

Extrahepatic manifestations of alveolar echinococcosis were excluded using cranial MRI and chest CT. The patient commenced with antihelmintic therapy with albendazole after transplantation and was discharged with overall improved health for rehabilitation. A CT of the liver six months after transplantation did not show any signs of echinococcus-induced lesions. Currently, 14 months after transplantation, the patient is in very good condition. Liver function remains normal, and both immunosuppressive treatment with tacrolimus (administered as prolonged-release advagraf, Astellas Pharma, Germany) and antihelmintic treatment with albendazole are well-tolerated.

## 2. Discussion

In the present case, the deterioration of an AIH/PSC patient was treated with orthotopic liver transplant, upon which the histology of the liver explant surprisingly revealed an infection by *Echinococcus multilocularis* that progressed locally, which was associated with elevated numbers of IgG4-positive plasma cells. Elevated IgG4-positive plasma cells were also detected in a bile duct biopsy two months prior to transplantation.

In our patient, there were no clear radiological signs of a parasitic infection, while the elevated ANA titer and the previous liver histology indicated AIH. Additionally, elevated transaminases corresponded to this diagnosis. The immunosuppressive medication for AIH might have possibly supported the exacerbation and atypical presentation of E. *multilocularis* infection prior to transplantation [2]. The strong involvement of the hilus region by Echinococcosis accompanied by the inflammatory destruction of a bile duct is striking since many metacestodes are associated with right sided bile ducts, leading to compression and stenosis.

At the time of transplantation, diagnostic criteria for PSC were fulfilled, and common causes for secondary sclerosing cholangitis were excluded.

The elevated IgG4 plasma cell levels in biopsy samples obtained from ERCP prior to transplantation supported the diagnosis of PSC since elevated IgG4 levels can occur in the serum or bile ducts of PSC patients [3,4]. Nevertheless, IgG4 plasma cell levels can be increased due to infection with parasites. In patients with AE, specific IgG4 antibodies are enriched [5], and a prominent IgG4 response has been associated with progressive parasitic disease [6]. In fact, IgG4 is the dominant IgG antibody isotype in AE [7,8,9]. To our knowledge, systematic histological evaluation of IgG4-positive plasma cells in resected hepatobiliary tissue in patients with AE has not been performed yet. Therefore, confronted with increased numbers of IgG4-positive plasma cells in biopsies, the pathologist should consider a parasitic infection as a differential diagnosis of PSC-IgG4.

Due to the fact that alveolar echinococcosis (AE) is endemic in Germany, the hepatologist should also include AE in the differential diagnosis in cases of pathological changes in the biliary tract and, thus, perform a serological test for Echinococcus multilocularis before liver transplantation.

Retrospectively, it appears most likely that alveolar echinococcosis had imitated the clinical picture of PSC, with all PSC-like findings caused by echinococcosis-induced bile duct involvement. However, we could not have excluded the possibility that the patient in our case had AIH, developed PSC and additionally had an infection with *Echinococcus multilocularis*.

## 3. Conclusions

Although a rare case, we suggest serological testing for echinococcosis to be performed in the case of ambiguous clinical and radiological findings in patients with chronic liver disease, especially in PSC patients without IBD. In general, physicians should consider AE in their differential diagnosis in regions where E. multilocularis is endemic.

It is well-known that AE may clinically and radiologically imitate carcinoma. To our knowledge, an imitation of PSC has not been reported yet. Apart from that, liver transplantation due to extensive AE remains a feasible therapeutic approach [10,11].

## Figures and Tables

**Figure 1 pathogens-12-00073-f001:**
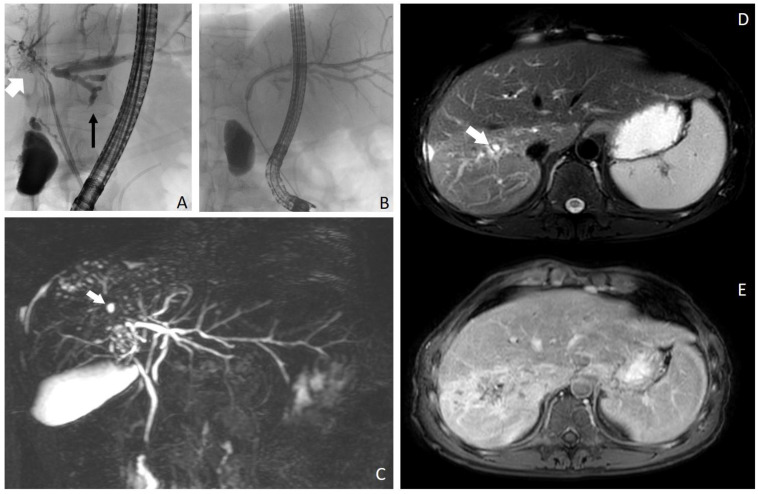
(**A**–**E**). ERCP revealed dominant stricture of the common biliary duct, atrophy of the right (white arrow) biliary system and hypertrophy of the left (black arrow) biliary system (**A**). Dilatation of the perihiliary left bile duct (**B**). In (**C**), maximum intensity projection of a 3D image using magnetic resonance cholangiopancreatography (MRCP), showing rarefication of the right sided bile ducts (especially segments VI and VII) with small cystic lesions (white arrow). MRI images show alterations of liver parenchyma in segments VI and VII. On fat-suppressed T2-weighted images (**D**), signal intensity increase is seen in segments VI and VII. The bile ducts are scarcely visible. Some small cystic lesions are seen in the corresponding segments (white arrow). These changes were interpreted as segmental changes due to primary sclerosing cholangitis. On contrast-enhanced fat-suppressed T1-weighted images (**E**), diffuse contrast enhancement is visible in segments VI, VII and VIII, which was attributed to inflammation.

**Figure 2 pathogens-12-00073-f002:**
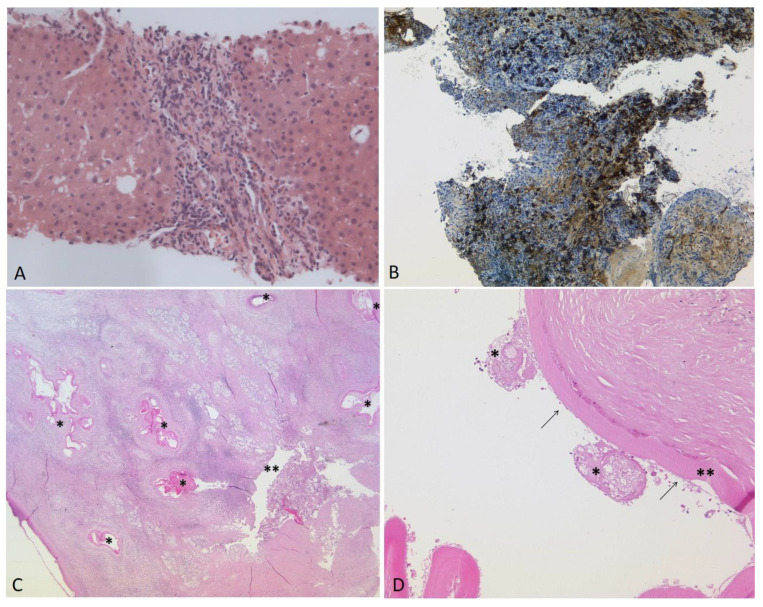
(**A**–**D**). (**A**): Liver biopsy three years before transplantation: Chronic-active, predominantly portal and discrete lobular inflammatory infiltrate with low activity and few eosinophilic granulocytes. No histological signs of infection by *E. multilocularis* were detected (HE, ×200). Several biopsies of the large bile ducts were taken during the months before transplantation. In the last biopsy ((**B**), ×100), two months before transplantation, large amounts of IgG4-positive plasma cells were visible (>80/HPF). Post-transplantation: (**C**): The histology of the explanted liver showing the laminated layer of *E. multilocularis* (*) in close proximity to a large bile duct (**) in the hepatic hilus region. The inflammatory reaction resulted in the compression and destruction of the bile duct (HE, ×12.5). (**D**): Magnification of a metacestode (HE, ×100). Remnants of two protoscolices (*), the laminated layer (**) and the delicate germinal layer (arrows) are shown.

**Figure 3 pathogens-12-00073-f003:**
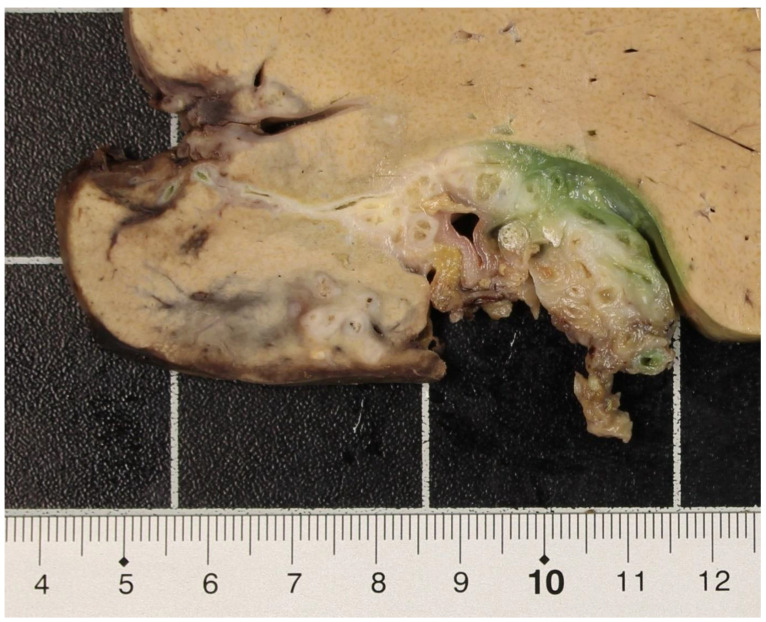
Hilar region of the liver explant with fibrotic strands and metacestodes.

**Figure 4 pathogens-12-00073-f004:**
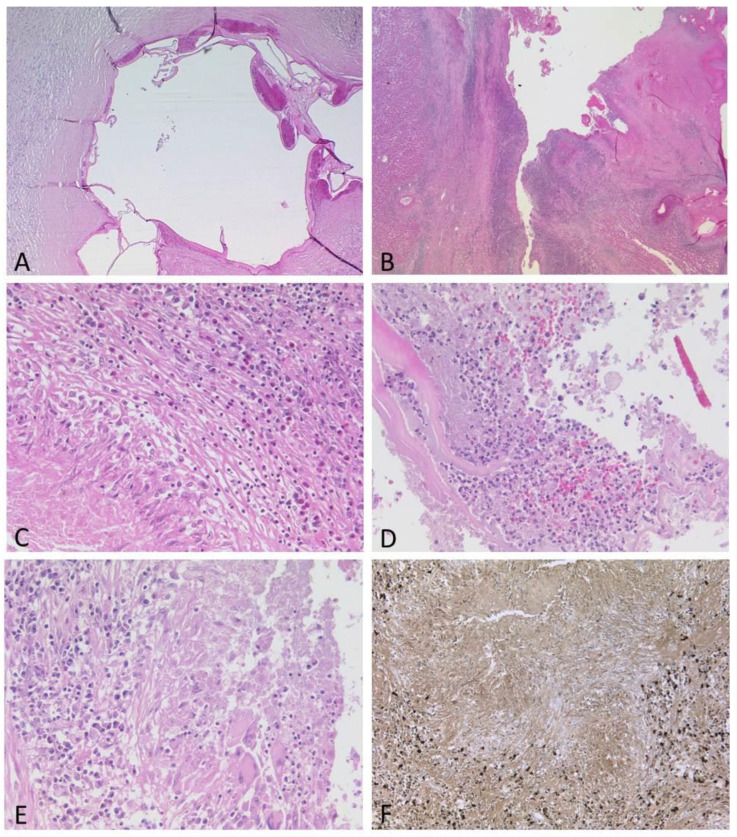
(**A**–**F**): Spectrum of histological features of the lesions. Some metacestodes contained calcifications, while the adjacent liver tissue showed severe fibrosis with loose lymphocytic infiltrate ((**A**), HE, ×40). Other metacestodes were largely destroyed by inflammation ((**B**), HE, ×12.5). Areas with granulomas and eosinophils ((**C**), HE, ×200), abscess formation ((**D**), HE, ×200) or numerous plasma cells ((**E**), HE, ×200), many of them IgG4-positive ((**F**), ×100), were present.

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
