# Peer review of "Alveolar Echinococcosis in a Patient with Presumed Autoimmune Hepatitis and Primary Sclerosing Cholangitis: An Unexpected Finding after Liver Transplantation"

_pathogens, 2023, doi:10.3390/pathogens12010073_

Round 1
Reviewer 1 Report
The submitted manuscript described a successful liver transplantation on a patient with presumed autoimmune hepatitis and primary sclerosing cholangitis, despite the fact that, alveolar echinococccosis had been diagnosed in a liver-explant. This is a valuable paper because an imitation of AE to PSC has not been reported yet. Moreover, as it was writtened a systematic histological evaluation of IgG4 positive plasma cells in resected hepatobiliary tissue in patients with AE has not been performed yet. Therefore, confronted with increased numbers of IgG4 positive plasma cells in biopsies, the pathologist should consider a parasitic infection as differential diagnosis to PSC-IgG4.
Article is well constructed, described and developed. Results are clearly presented. However, there is lack of summary/conclusion of the manuscript. There are a few stylistic and punctuation errors and I suggest that it need some minor corrections and text editing before publication.
Please find the rest of comments and suggestions in the pdf file.

Author Response
Article is well constructed, described and developed. Results are clearly presented. However, there is lack of summary/conclusion of the manuscript. There are a few stylistic and punctuation errors and I suggest that it need some minor corrections and text editing before publication.
Please find the rest of comments and suggestions in the pdf file.
--> We adopted all your suggestions and correction. There is now a short conclusion at the end oft he report.
Reviewer 2 Report
Should be included in the discussion/conclusion: Due to the fact that alveolar echinococcosis (AE) is endemic in Germany this disease should be included in differential diagnosis in any case of suspected pathologic changes in the liver and the biliary tract and thus, a specific Echinococcus multilocularis serological test should be carried out before liver transplantation. and why this aspect was not considered.
Author Response
Should be included in the discussion/conclusion: Due to the fact that alveolar echinococcosis (AE) is endemic in Germany this disease should be included in differential diagnosis in any case of suspected pathologic changes in the liver and the biliary tract and thus, a specific Echinococcus multilocularis serological test should be carried out before liver transplantation and why this aspect was not considered.
--> In our patient, there were no clear radiologic signs of a parasitic infection, while the elevated ANA-titer and previous liver histology indicated AIH. Additionally, elevated transaminases corresponded to this diagnosis. Since these findings seemed conclusive, Echinococcus was not considered as differential diagnosis.
Reviewer 3 Report
Pathogens-2086458: Alveolar Echinococcosis in a patient with presumed autoimmune hepatitis and primary sclerosing cholangitis: An unexpected finding after liver transplantation
General comments:
AE is a parasitic disease with zoonotic character that can be fatal if untreated.
In this case report, authors report on a patient with presumed autoimmune hepatitis and primary sclerosing cholangitis. After liver transplantation, alveolar echinococcosis was diagnosed in the explant-liver.
It is very valuable in terms of revealing the fact that this disease should not be ignored in people in regions and countries where E. multilocularis is endemic (such as Germany).
Specific comments:
Page 4, L114-123: In Figure 2 A-B, the time is indicated as before transplantation, but not in C-D. I think it's post-transplantation, if so it should be added to the text.
Page 4, L119: … layer of E. multilocularis (*) in…
Page 6, L148: tacrolimus (advagraf, add brand, country)
Page 7, L185: A sentence that will create awareness can be added here, such as “physicians should consider AE in regions where E. multilocularis is endemic.
Author Response
Specific comments:
Page 4, L114-123: In Figure 2 A-B, the time is indicated as before transplantation, but not in C-D. I think it's post-transplantation, if so it should be added to the text.
--> We habe adopted your suggestion.
Page 4, L119: … layer of E. multilocularis (*) in…
--> We have corrected that part.
Page 6, L148: tacrolimus (advagraf, add brand, country)
--> administered as advagraf with prolonged- release, Astellas Pharma, Germany
Page 7, L185: A sentence that will create awareness can be added here, such as “physicians should consider AE in regions where E.
multilocularis is endemic
--> . Following sentence was included in the conclusion: "In general, physicians should consider AE in their differential diagnosis in regions where E. multilocularis is endemic."